# Superparamagnetic Hybrid Nanospheres Based on Chitosan Obtained by Double Crosslinking in a Reverse Emulsion for Cancer Treatment

**DOI:** 10.3390/polym15234493

**Published:** 2023-11-22

**Authors:** Mohammed Dellali, Kheira Zanoune, Mihaela Hamcerencu, Corina-Lenuța Logigan, Marcel Popa, Hacene Mahmoudi

**Affiliations:** 1Faculty of Technology, University Hassiba Benbouali of Chlef, Chlef BP 151 02000, Algeria; m.dellali@univ-chlef.dz (M.D.); k.zanoune@univ-chlef.dz (K.Z.); h.mahmoudi@univ-chlef.dz (H.M.); 2Laboratory of Natural Bio-Resources, University Hassiba Benbouali of Chlef, Chlef BP 151 02000, Algeria; 3CQFD Composites, Village Industriel de la Fonderie, François Spoerry Street, No. 65, 68100 Mulhouse, France; mhamcerencu@yahoo.com; 4Department of Natural and Synthetic Polymers, Gheorghe Asachi Technical University of Iasi, Bld. Prof. Dr. Doc. Dimitrie Mangeron Street, No. 73, 700050 Iasi, Romania; 5Faculty of Medical Dentistry, “Apollonia” University of Iasi, Pacurari Street, No. 11, 700511 Iasi, Romania; 6Academy of Romanian Scientists, Ilfov Street, No. 3, Sector 5, 050094 Bucharest, Romania

**Keywords:** chitosan, superparamagnetic hybrid nanospheres, double crosslinking, reverse emulsion, cancer treatment

## Abstract

Nowadays, the Magnetically Targeted Drug Delivery System (MTDDS) is among the most attractive and promising strategies for delivering drugs to the target site. The present study aimed to obtain a biopolymer–magnetite–drug nanosystem via a double crosslinking (ionic and covalent) technique in reverse emulsion, which ensures the mechanical stability of the polymer support in the form of original hybrid nanospheres (NSMs) loaded with biologically active principles (the 5-Fluorouracil (5-FU)) as a potential treatment for cancer. Obtained NSMs were characterized in terms of structure (FT-IR), size (DLS), morphology (SEM), swelling, and 5-FU entrapment/release properties, which were dependent on the synthesis parameters (polymer concentration, dispersion speed, and amount of ionic crosslinking agent). SEM analysis results revealed that NSMs presented a spherical shape and are homogeneous and separated. Moreover, NSMs’ ability to load/release 5-FU was tested in vitro, the results confirming, as expected, their dependence on the varied synthesis process and NSM swelling ability in physiological liquids. The drug transport mechanism through the polymer matrix of its release is the Fickian type. The morphological, bio-material characteristics and the ability to include and release an antitumor drug highlight the utility of the NSMs obtained for targeting and treating some tumor diseases.

## 1. Introduction

One of the most popular diseases that cause death worldwide is cancer [1]; it is a group of diseases that can affect any part of the body. It is characterized by the rapid multiplication of abnormal cells with unusual growth, which then invade other parts of the body. When these abnormal cells form clumps in vital organs, it leads directly to death. Despite the wide range of available drugs in the market, there are still many unsatisfactory treatments. To meet these needs, one of the research priorities worldwide is the need to discover new treatments for varied types of cancers; thus, considerable efforts are being made to find a proficient treatment strategy and achieve a significant improvement in the medical service.

The application of nanotechnology in oncology presents many advantages [2], such as paving the way for various therapies to eliminate cancer cells and the possibility of delivering drugs to the tumor site with greater specificity. Nanotechnology presents the potential to produce completely new and highly effective therapeutic agents. Advances in the use of nanotechnology have made drugs more “intelligent”.

Nowadays, a Magnetically Targeted Drug Delivery System (MTDDS) is among the most attractive and promising strategies to deliver drugs to the target site. Compared to traditional techniques, MTDDS is very effective and has a rapid impact; additionally, it can reduce toxicity and unwanted side effects [3]. Targeted delivery systems hold great potential in cancer therapy due to specific drug delivery to cancerous tissues. Magnetic nanoparticles represent the key compound in these administration systems thanks to their magnetic properties: they can be selectively targeted to specific (cancerous) tissues in the presence of a magnetic field. Magnetic nanoparticles (NPs) consist of magnetic elements, such as iron, nickel, cobalt, chromium, manganese, gadolinium, and their chemical compounds. Different studies show that the magnetic NPs based on ferrite nanoparticles are the most explored [4]. The NPs based on ferrite nanoparticles are endowed with properties such as narrow size distribution, good dispersibility and biocompatibility, reduced toxicity, superparamagnetism, and high chemical stability under physiological conditions. All these unique advantages mentioned make them suitable candidates as multifunctional nanocarriers [5]. Various studies revealed the use of magnetic NPs as heat generators, which can induce localized cell death [6]. Based on these findings, researchers succeeded in combining chemotherapy and thermotherapy to achieve an effective cancer treatment [7]. These magnetic NPs cannot be used alone as vectors of drugs [8] due to their instabilities in aqueous solutions, their aggregations, and their precipitations.

On the other hand, polymeric NPs are the main branch of nanotechnology that can be used for diagnostic and therapeutic purposes. Polymers are widely used in the preparation of particulate systems and occupy an important place in this field. NPs based on biopolymers have been extensively used as a support for drug carriers due to their significant advantages for efficient delivery [9,10]. There are two structural types of polymeric nanoparticles: nanocapsules and nanospheres. During this study, the experimental part was focused on nanospheres as nanoparticles with a spherical shape and smooth texture, which are adequate for delivering anticancer agents to solid tumors and directing them easily through the tumor vasculature and into tumor cells [11].

Chitosan (CS) is one of the biopolymers that can form NPs with unique properties [12,13]. CS received great interest for applications in the medical and pharmaceutical field due to its attractive properties like biocompatibility, biodegradability, and non-toxicity. It is also known for other properties, such as analgesic, mucoadhesiveness, hemostatic, non-antigenicity, hydrophilicity, antimicrobial activity, and antioxidant [14,15,16,17]. CS has an intrinsic antitumor property, which makes it an ideal vector for the treatment of cancer [18]. CS-based NPs, due to their simple preparation process, are of great interest and are widely exploited due to their suitability as promising vectors in the pharmaceutical industry for controlled drug delivery [19].

Hybrid NPs obtained by combining magnetic NPs with a natural polymer matrix hold promise for advanced biomedical applications: bio-detection and bioseparation, diagnostics, magnetic resonance imaging, hyperthermia therapy, and controlled drug delivery. The combination of magnetic properties with various polymer properties leads to new systems with specific functionalities. The magnetic properties of these materials give the vectors a functionality that opens up prospects for new, original therapeutic applications (such as targeting via magnetic guidance and magneto cytolysis induced by hyperthermia) and diagnostics (such as contrast in magnetic resonance imaging (MRI)).

Various efforts have been devoted to developing different types of drug delivery systems capable of delivering anticancer agents directly to the desired site via sophisticated targeting strategies. The method for preparing NPs must be based on the desired properties, such as particle size, thermal and chemical stability, the nature of the active compound, reproducibility, and reduced toxicity.

This study aimed to obtain a biopolymer–magnetite–drug nanosystem in the form of original hybrid NPs loaded with biologically active principles as a potential treatment for cancer. The prepared NPs are based on natural polymer (CS), biocompatible, in the form of nanospheres containing both magnetic nanoparticles of iron oxide (magnetite) dispersed in their matrix and an anticancer drug, 5-Fluorouracil (5-FU). Further, the hybrid NPs may present two potential applications: it can be directed in particular using an external magnetic field towards the target site, and on the other hand, the application of an alternating external magnetic field can lead to an increase in temperature in the tumor cells, inducing a hyperthermic effect that will destroy tumor cells. The innovation of the present study is the technique used for obtaining these particles.

The work presented herein is focused on obtaining biocompatible magnetic nano-spheres (NSMs) based on crude CS, which allow the inclusion of magnetite and, at the same time, the loading of biologically active substances suitable for the treatment of certain cancer diseases. For the preparation of hybrid nanoparticles, a double crosslinking, ionic, followed by a covalent, reverse-emulsion (*w*/*o*) procedure was used. A tripolyphosphate solution was used for the ionic crosslinking of the polymer, and glutaraldehyde was used as the covalent crosslinking agent. Double crosslinking was performed to maintain the stability of the polymer particles without an increase in the toxicity of the particulate system. NSMs were characterized via FT-IR, DLS, and SEM, and the water uptake ability and drug loading/release capacity were tested. The analysis result confirmed that NSM properties are dependent on the used synthesis parameters.

## 2. Materials and Methods

### 2.1. Materials

Chitosan (CS) (low molecular weight degree of deacetylation 82%), acetic acid (99%), glutaraldehyde (GA) (25% aqueous solution), tripolyphosphate (TPP), n-hexane, toluene, acetone, surfactants (Tween 80, Span 80), and 5-Fluorouracil (5-FU) were purchased from Sigma-Aldrich. Sodium hydroxide (NaOH) and iron (III) chloride anhydrous were purchased from Lachner. Iron (II) chloride tetrahydrate (FeCl_2_·4H_2_O), acetate, and phosphate-buffer saline solutions (ABS and PBS) were purchased from Fluka. Double-distilled water (DDW) was freshly produced in our laboratory. All other reagents used in this study were of analytical-grade purity and were used without further purification.

### 2.2. NSMs Preparation Methods

#### 2.2.1. Magnetite Nanoparticle Preparation

In this study, the magnetite NPs were obtained via the co-precipitation method described by Hritcu et al. [20], with slight modifications. This chemical method is most frequently used due to its simplicity, controllable handling, and efficiency [21]. The co-precipitation reaction was performed in a closed flask in an inert atmosphere in a basic medium. Thus, 0.0275 moles of iron (II) chloride tetrahydrate FeCl_2_·4H_2_O was dissolved in a certain volume of distilled water (84 mL). Separately, 0.055 moles of iron (III) chloride FeCl_3_ was dissolved in a glass flask containing 90 mL of distilled water. Both solutions were kept under magnetic stirring until complete dissolution. Further, a non-ionic surfactant solution (36 mL, 2% *w*/*w*) Pluronic F127 was added to each solution. Next, in a glass flask (500 mL), the two solutions were mixed and placed into a heating bath equipped with magnetic stirring, preheated at a temperature of 70 °C. To prevent the premature oxidation of iron (II) in iron (III), the solution was maintained under N_2_ gas. Further, to the reaction flask, a NaOH solution (12.8 g in 120 mL H_2_O) was added dropwise. The reaction was maintained and left to react for another 30 min. The resulting precipitate was purified by washing with DDW several times to reduce the pH from 13 to 7 and to remove the unreacted precursors and products. Obtained NPs were recovered using a strong magnet applied to the wall of the flask. NPs were dried in a vacuum oven at 50 °C for 24 h. Next, NPs were kept at 50 °C in an oven until a constant mass of Fe_3_O_4_ was attained.

#### 2.2.2. Preparation Method of Magnetic Nanospheres (NSM)

NSMs were obtained using a double crosslinking (ionic and covalent) technique in a reverse emulsion (*w*/*o*) according to a method reported by Peptu et al. [22,23,24,25].

Initially, several aqueous polymer solutions were prepared by dissolving a certain amount of CS (depending on the desired concentration) (Table 1) in an acetic acid solution (1% *w*/*w*) under magnetic stirring. In this solution, a certain volume (10 mL) of magnetite aqueous suspension was added under continuous stirring. An appropriate quantity of nonionic surfactant Tween 80 (2% *w*/*w*) was added to the polymer solution as a hydrophilic stabilizing agent. Further, the mix was ultrasonicated. Separately, the organic phase was prepared as follows: in a 600 mL beaker, 200 mL toluene and an adequate amount of surfactant, Span 80 (2% *w*/*w*), were added. Aliquots (100 μL) of the mix were added into the organic phase under Ultra-turrax stirring (5000–15,000 rotations per minute, (rpm)), leading to emulsion formation. Subsequently, during the emulsion stabilization time, freshly ionic crosslinker solution (5% *w*/*w* TPP) was added slowly in drops to the formed *w/o* emulsion. After 15 min, the mixture was transferred to a mechanical stirring-equipped reactor, where the process of ionic crosslinking continued at 500 rpm. The covalent crosslinking process was carried out by adding a calculated amount of glutaraldehyde previously extracted from toluene (c = 1.12 mg/mL). After 1 h, the crosslinking process was completed, and the suspension containing magnetic particles was centrifuged for 1 h at 15,000 rpm to separate them from the supernatant. Finally, purification by successive washings with double-distilled water, acetone, and n-hexane to remove TPP and GA surfactants, respectively, was performed. The purified NSMs were characterized to determine their composition (FTIR and TGA), magnetic properties, size distribution, and morphology via scanning electron microscopy (SEM); Zeta Potential; swelling behavior in an aqueous environment; drug loading; and in vitro drug release.

For the synthesis, the following variables were taken into account: the amount of magnetite added to the CS solution (polymer/magnetite *w*/*w* ratio), the polymer solution concentration, speed agitation on Ultra-turrax, and the molar ratio between the polymer amino groups and ionic crosslinker. The experimental protocol with the varied parameters used in obtaining hybrid nanospheres is presented in Table 1.

### 2.3. NSM Characterization

#### 2.3.1. Fourier Transform Infrared Spectroscopy (FT-IR)

In this study, Fourier Transform Infrared Spectroscopy (FT-IR) was used to confirm the crosslinking process and to visualize the presence of the magnetite in the prepared NSMs. FT-IR analysis was performed with a Digilab Scimitar FTS 2000 FTIR spectrometer in the transmittance mode ranging from 450 to 4000 cm^−1^ at a resolution of 4 cm^−1^ using the pellet procedure with KBr.

#### 2.3.2. Scanning Electron Microscopy (SEM)

The morphological analysis of the NSMs (size, shape, and surface morphology) was investigated via SEM technique. SEM images were performed using a HITACHI SU 1510 (Hitachi SU-1510, Hitachi Company, Tokyo, Japan) Scanning Electron Microscope. Before observation, NSMs were fixed on an Aluminum stub and coated with a 7 nm thick gold layer using a Cressington 108 device.

#### 2.3.3. Dynamic Light Scattering (DLS) and Zeta Potential

The mean diameter of hybrid nanoparticles, the size distribution, and Zeta potential were determined in triplicate at 25 °C at a concentration of 1% (*w*/*v*) using a Zeta Nanosizer Malvern (Malvern, UK). The mean diameter and size distribution were determined for samples dispersed in anhydrous acetone to avoid the swelling of the particles. Nanosphere Zeta potential was determined by electrophoresis in a phosphate-buffered solution (PBS; pH = 7.4).

#### 2.3.4. Thermal Analysis (TGA)

The thermogravimetric analysis allows us to characterize the composition of materials by measuring their weight loss as a function of temperature. TGA measurements of NSMs were performed with a TA Instrument Q600 analyzer in an air atmosphere (100 mL/min) at a heating rate of 10 °C/min from ambient temperature to 700 °C. The samples weighed between 8 and 10 mg, and the measurement parameters were kept constant during all experiments to obtain comparable data.

#### 2.3.5. NSM Magnetic Properties

NSMs’ magnetic properties (the saturation magnetization—Ms, the remanent magnetization—Mr, and Coercivity—Hc) were evaluated by vibrating sample magnetometry method using a vibrating sample magnetometer (VSM) (MicroMag, Model 3900, Princeton Measurements Corporation, Princeton, NJ, USA) on dried powders at room temperature.

#### 2.3.6. NSM Swelling Ability in Aqueous Media

The current study aims to use these hybrid nanospheres as drug carriers; therefore, the swelling properties in aqueous solutions are important, thus influencing the NPs’ drug loading and release capacity. The gravimetric method was used to evaluate swelling behavior using buffer solutions with different pHs. A total of 30 mg of dried NSMs was placed in an Eppendorf tube containing 1 mL acetate buffer solution (ABS) (pH = 3.6), in an aqueous medium with a pH = 6.7, or phosphate-buffered solution (PBS) (pH = 7.4). The obtained suspension was maintained at room temperature under magnetic stirring for 24 h. At specific time intervals, the NSM suspensions were ultracentrifuged (15,000 rpm), the supernatant and the excess liquid were removed by carefully wiping with filter paper, and the swollen sample was weighed. All experiments were performed in triplicates. The water-uptake ratio (Q%) was calculated with Equation (1):(1)$$Q\left(\%\right)=\frac{W-W_{0}}{W_{0}}\cdot 100$$
where

W: the weight of the swollen sample (mg).W_0_: the initial weight of the dry sample (mg).

#### 2.3.7. NSMs’ 5-FU Loading Capacity

The 5-FU-loading process was performed using diffusion mechanism. To evaluate the loading efficiency of the drug in the obtained hybrid nanoparticles, we followed the hereafter experimental steps. Each sample of a specific amount of magnetic nanospheres (0.03 g) was accurately weighed and dispersed in a 1.5 mL aqueous 5-FU solution (10 mg/mL). The suspension was maintained for 24 h under gentle mechanical stirring (50 rpm) at room temperature. After preset times, samples were ultracentrifuged at 15,000 rpm for 5 min. The quantity of 5-FU loaded into NSMs was determined by the difference between the initial 5-FU quantity and the 5-FU quantity from the supernatant after ultracentrifugation based on a calibration curve previously obtained. The measurements were performed using a spectrophotometer UV–VIS Nanodrop ND 1000 at 266 nm, which allows the analysis of very small sample volumes (0.2 µmL). The 5-FU loading efficiency (Lef %) into NSMs was calculated as follows:(2)$$m_{1}=m_{i}-m_{s}$$
(3)$$L_{ef}\left(\%\right)=\frac{m_{i}-m_{s}}{m_{i}}\cdot 100$$
where
m_l_: the amount of loaded 5-FU (mg).m_i_: the initial amount of 5-FU (mg).m_s_: the amount of 5-FU found in supernatant (mg).

The entrapped drug was calculated using a calibration curve of 5-FU at 266 nm, previously performed, with Equation (4):(4)$$Y=5.141x;R^{2}=0.9971$$

#### 2.3.8. NSMs’ 5-FU Release Kinetics

The 5-FU-release process was performed in two different mediums: at pH = 7.4, similar to blood’s pH, and at pH = 6.7, similar to a tumor area. In vitro, 5-FU-release studies were performed using the dialysis method. Briefly, individually lyophilized samples of 5-FU-loaded magnetic nanospheres were introduced into the dialysis membrane (MWCO 14,000, Ø 22 mm) and immersed in glass flasks containing a well-known PBS volume (13 mL. All NSM samples were maintained at 37 °C ± 0.5 °C under continuous gentle mechanical stirring during the release period. The amount of released 5-FU was quantified spectrophotometrically at a wavelength of 266 nm. The release efficiency of 5-FU (R_ef_ %) was determined using Equation (5):(5)$$R_{ef}\left(\%\right)=\frac{m_{r}}{m_{l}}\cdot 100$$
where
m_r_: 5-FU released from NSMs (mg).m_l_: 5-FU loaded into the NSMs (mg).

#### 2.3.9. Theoretical Analysis of 5-FU Release

The purpose of the application of theoretical mathematical models is the study of the particular drug-release process of various therapeutic systems. Subsequently, mathematical models have been used to design several simple and/or complex drug-delivery systems, respectively, to predict the overall release behavior. In 1983, Korsmeyer and Peppas developed a semi-empirical equation to describe the drug-release process from a polymer system [26]:(6)$$\frac{M_{t}}{M_{\infty }}=k\cdot t^{n}$$
where
t: release time.M_t_: drug amount delivered at time t.M∞: total amount of loaded drug.k: kinetic constant, which is a measure of the release rate.n: is the diffusional exponent, which indicates the mechanism of the drug release.

## 3. Results and Discussion

### 3.1. Preparation and Characterization of NSMs

The formation of CS-based magnetic nanospheres was performed in a weakly acidic medium, where the amine groups of chitosan ionize and transform into ammonium cations, which can interact with the anions present in the structure of the crosslinker.

The ionic crosslinking reaction occurs between positively charged CS (NH^3+^) and negatively charged tripolyphosphate (P_3_O_10_^5−^), TPP being electrostatically attracted to the NH^3+^ groups of CS, leading to ionically crosslinked CS nanoparticles. In addition, GA reacts with the polymer, mainly on the surface layers of the ionically crosslinked particles, thus giving rise to hard nanoparticles [27,28]. The covalent crosslinking process takes place between the carbonyl functions of GA and the amine groups of CS with the formation of imine-type bonds via the basic Schiff reaction. In addition, GA reacts with the polymer, mainly on the surface layers of the ionically crosslinked particles, thus giving rise to hard nanoparticles [28]. The magnetic NPs dispersed in the polymer matrix can be located either in the center or on the surface of the NSMs. Hydrogen bonds are also formed by the –NH^3+^ groups of the CS, which are attracted by the –OH functions of the iron oxide [29]. The schematic structure of the obtained NSMs is shown in Figure 1, with the presence of magnetic NPs being noticed both in the core of the NSMs and in the polymer membrane.

The influence of certain synthesis parameters on final NSMs properties was studied, namely, the CS/Fe_3_O_4_ mass ratio, CS initial concentration, NH_2_/TPP molar ratio, and stirring speed in the emulsion formation phase. Other parameters, such as the initial water/oil phase ratio, surfactant concentration, and GA content in the organic phase, were kept constant throughout the study. The experimental program containing sample codes and experimental conditions is presented in Table 1. For all samples, the following parameters have been used: –NH_2_/GA = 1/0.5 molar ratio; the ionic crosslinking time is equal to 60 min and 90 min for covalent crosslinking.

FT-IR spectroscopy was used to confirm the ionic and covalent bonding and the presence of magnetic material in the newly formed nanospheres.

The recorded spectra revealed similar profiles for all NSMs, with a slight difference in the adsorption intensities of some bands due to their environment. The results, therefore, confirm the success of the double-crosslinking process. Moreover, all the characteristic CS and iron oxide peaks are present in the spectra of the magnetic nanospheres. For this reason, only FTIR spectra for samples NSM-1, NSM-2, NSM-10, crude CS, magnetite (Fe_3_O_4_), and non-magnetic nanospheres (NS) are presented (Figure 2).

The recorded FTIR spectra for crude CS (Figure 2) showed the absorption bands at 1076 cm^−1^ (C–O–C stretching vibration) [28], NHAc units, amide I, NH2 bending, and amide III, respectively, at 1659 cm^−1^, 1423 cm^−1^, and 1378 cm^−1^ [29]. A strong peak at 3470 cm^−1^ attributed to the axial stretching vibration of (OH) overlapped with the (NH_2_) stretching band of CS molecules was also noticed [30,31].

The hybrid nanospheres’ spectra revealed an absorption band at 1647 cm^−1^, indicating the C=N stretching vibration of the imine group of the Schiff base; this denotes a clear indication of covalent crosslinking between the amine groups of CS and the carbonyl groups of glutaraldehyde. Moreover, the absorption band signal at 893 cm^−1^ is specific to newly formed bonds, proving the ionic crosslinking between TPP (–P–O–P–) anions and CS ammonium cations [25].

The presence of the magnetic material in the structure of the composite nanospheres is confirmed by the superposition of the peaks at a signal of 580 cm^−1^ corresponding to the Fe–O/Fe–O–Fe limits of magnetite (Fe_3_O_4_) [25,32,33].

### 3.2. Mean Diameter Determination

Considering the potential application of the magnetic nanospheres (NSMs), namely, the administration as an injectable suspension into the bloodstream near tumors, size is an important parameter. Particle-size measurements were performed via dynamic light scattering and confirmed that magnetic NPs have an average diameter of 14 nm, and the size of the hybrid nanosphere varied between 190 and 531 nm (Figure 3). The results also showed that the particle size is dependent on the synthesis parameters (magnetite quantity, polymer concentration, dispersion speed, and amount of ionic crosslinking agent).

Also, the experimental data confirmed that the size of the NSMs depends on the concentration of the polymer solution and the stirring speed used during the preparation. The decrease in the concentration of CS in the solution leads to the formation of smaller particles (Figure 3b, the case of sample NSM-1, which presents the smallest size and polydispersity). Increasing the CS concentration led to the formation of larger particles due to the higher viscosity of the solution and the formation of bigger droplets during emulsification, which subsequently hardened in the presence of crosslinking agents (sample NSM-7).

As stated previously, a higher stirring speed leads to obtaining smaller particles. This phenomenon can be explained by the fact that a higher stirring speed implies the dissipation of greater energy in the stirred emulsion, which means a greater energy transfer, which causes fragmentation of the emulsion into smaller droplets, thus giving rise to nanometric particles (Figure 3c). So, for samples NSM-1, NSM-10, and NSM-11, after increasing the stirring speed from 12,000 to 18,000 rpm, it was observed that the size decreases due to better dispersion and the formation of small droplets in the emulsion (as, for example, is the case for NSM-11). The dynamic light scattering curves for all analyzed samples have a monomodal appearance, which demonstrates that the hybrid NPs have a homogeneous size and an average nanometer diameter depending on the different preparation parameters.

### 3.3. NSM Zeta Potential

Zeta potential analysis was performed to investigate the aqueous suspension stability of NSMs in a slightly alkaline medium at pH = 7.4. The results obtained are highlighted in Table 2, and were evaluated based on the Smoluchowscki equation and ranged from −8.15 to −19.7 mV. Published studies on colloid stability confirmed that colloidal dispersions are thermodynamically unstable if particle Zeta potential values are not in the 25–30 mV range [34]. Based on the results obtained, we conclude that NSMs present lower stability in an aqueous medium. Even though the stability of NSM systems is lower over time, we state that the system can still be used for intravenous administration if the injection is performed within a few minutes after preparation of the suspension.

### 3.4. Hybrid Nanoparticle Morphology

The SEM technique was used to analyze the morphology of NSMs. SEM pictures of the most representative samples are shown in Figure 4, Figure 5 and Figure 6. The results indicate that NSMs exhibit a spherical shape and are well individualized, generally with a submicron diameter and reduced dimensional polydispersity. We can also notice that the particles are homogeneous and separated, without forming agglomerates. This statement is also supported by the fact that by resuspending in an aqueous medium, the dimensional polydispersity curves recorded for two randomly selected samples (NSM1 and NSM3) are practically similar to those shown in Figure 3 (data not represented in the manuscript).

The SEM images confirm the results obtained by dynamic light scattering, namely, that the preparation parameters influence the morphology of the NSMs. We observe, in particular, that a lower concentration of CS leads to obtaining the lowest dimensions of NSMs. For example, for the NSM-6 sample (Figure 5), obtained starting from a polymer solution diluted to 0.3%, we observe that the shape of the NS is irregular with the appearance of aggregates. This can be interpreted by the fact that at too high dilutions of the polymer solution, the crosslinking bridges between the amine groups of chitosan and the functional groups of the crosslinkers (ionic and covalent) become difficult to form, thus obtaining a low crosslinking degree.

In the case of NSM samples obtained with different CS/Fe_3_O_4_ weight ratios (Figure 4), it was observed that increasing the amount of magnetite incorporated in the hybrid NSMs led to the formation of aggregates. A possible explanation can be given by the presence of magnetite, in excess, not included in the polymer matrix, which destabilizes the particles. For magnetic nanospheres that contain less magnetite, particularly in the case of NSM-1 and NSM-2 with the weight ratio CS/Fe_3_O_4_: 1/0.5 and 1/0.7, we note that they present well-individualized spherical shapes.

Figure 6 confirms that the stirring speed is a decisive parameter for establishing the diameter of the nanoparticles. Thus, increasing the stirring speed from 5000 to 18,000 rpm leads to smaller particle sizes.

### 3.5. Thermal Behavior

Thermogravimetric analysis (TGA) of the NSMs was carried out for two purposes: on the one hand, to evaluate the magnetite content incorporated in the polymer matrix (according to the polymer/magnetite ratio used in the obtaining process) and, on the other hand, to simulate the thermal sterilization process in the case of materials for biomedical applications.

To evaluate the magnetite content in the polymer matrix, the study of thermal degradation was carried out in the temperature range between 25 and 700 °C with a speed of 10 °C/min under a nitrogen atmosphere. The results showed that all analyzed samples presented similar profiles, which is why Figure 7 presents the TGA curves of only two samples (NSM-1 and NSM-3).

In Figure 7, it can be observed that the thermal curves show three stages of degradation with different percentages of weight loss. The first stage of degradation takes place in the temperature range of 25 to 150 °C when there is a weight loss of 12 to 16%, characteristic of the evaporation of free water molecules. The second stage occurs up to a temperature of 345 °C (the particles begin to decompose at 200 °C) with a loss of 23 to 27% of their weight; this corresponds to the degradation of the saccharide rings and the disintegration of the macromolecular chains of the CS. The last stage of degradation is observed between 450 °C and 800 °C. It causes a weight loss of 18 to 21%, which corresponds to the complete decomposition of the polymer matrix.

It can be stated that the weight loss of the NSMs in the degradation process is mainly due to the degradation of the polymer, and the residue is composed of magnetic material. There is a logical sequence to all thermal curves; for sample NSM-1, we noticed a mass percentage of 42.49%, which remains at 790 °C, corresponding to the quantity of magnetite incorporated in the hybrid nanoparticles. Also, the percentage remaining in the NSM-3 sample was approximately 45.39%; this difference is linked to the initial CS/magnetite mass ratio incorporated in the hybrid nanoparticles (Table 1). Thus, magnetic nanospheres are thermally more stable than non-magnetic particles, with their thermal stability depending on the magnetite content. The results obtained are in good agreement with other studies [22,35].

### 3.6. Magnetic Properties

Figure 8 represents the magnetization curves and the saturation magnetization (Ms) of the NSMs (obtained at different CS/magnetite ratios) for NSM-1, NSM-2, NSM-3, and NSM-5. These results were determined by using a vibrating sample magnetometer (VSM) at room temperature. The magnetic response is directly proportional to the saturation magnetization (MS) of the material.

For all analyzed samples, the magnetization curves do not exhibit hysteresis when an external magnetic field is applied. The coercivity (Hc) and remanence (Mr) are zero, which proves the superparamagnetic properties of the obtained hybrid nanoparticles [22,36,37]. The concept is explained by the fact that when an external magnetic field is applied, the nanoparticles become magnetic, while returning to the non-magnetic state once the magnetic field stops. This property is essential for the application of this type of nanoparticles in the biomedical and bioengineering fields. The characteristic of superparamagnetism makes possible the applications of NSMs in targeted drug delivery by preventing their “active” behavior once the magnetic field is deactivated; the nanoparticles can then be eliminated from the body [38].

Saturation magnetization (Ms) values were recorded for each sample: for sample NSM-1, Ms = 10.47 emu/g; for NSM-2, Ms = 11.6 emu/g; for sample NSM-3, Ms = 15.07 emu/g; and for sample NSM-5, Ms = 25.84 emu/g. In Figure 8, it was observed, as expected, that the saturation magnetization increases when increasing the amount of magnetite used for preparation (with an increasing CS/Fe_3_O_4_ ratio). The smallest value of magnetization was recorded for the NSM-1, and the highest magnetization value was recorded for the NSM-5. Given the results, the analysis of the saturation magnetization values reinforces the idea that magnetic nanoparticles were successfully incorporated into CS-based nanoparticles.

### 3.7. Swelling Studies

To better understand the interaction between NSMs and the aqueous environment, swelling studies were performed in different aqueous solutions at different pH values (ABS: pH = 3.6 and 6.7, and PBS: pH = 7.4). The different pH values were chosen similarly to the suspensions intended to be used in biological fluids. The swelling of the magnetic nanospheres or the water-retention capacity in the nanospheres is caused by the presence of hydrophilic groups—in our case, the amino and hydroxyl groups of the polymer chains [39]. The swelling degree is influenced by the preparation parameters of NSMs and depends on the internal architecture of the newly formed polymer network and many other factors, such as the hydrophilicity of the materials, the amount of magnetic material in the nanospheres, and the crosslinking degree.

The swelling capacity of the nanospheres is generally caused by the penetration of water into the mesh of the polymer matrix. The swelling kinetics were studied over 24 h (equilibrium time) for all samples. The maximum values of the degree of swelling (Qmax), obtained at 24 h in different aqueous media at different pHs are presented in Figure 9. The obtained kinetic curves showed a profile identical to that presented in Figure 9.

The analysis of the results illustrated in Figure 9 shows that the nanospheres swell rapidly in the first minutes (30 min), followed by a slow increase for about 3–4 h; then, an equilibrium is established so that the maximum is reached at 24 h. For all the magnetic nanospheres, the maximum value of the swelling rate is reached around 200 min (after about two hours) due to the highly hydrophilic character of CS.

As expected, the swelling degree is influenced by preparation parameters, such as the polymer/magnetite ratio, polymer solution concentration, stirring speed, and polymer/crosslinker molar ratio. The results showed that the swelling degree also depends on the pH of the aqueous solution used for the study. The analysis of the swelling behavior led to interesting results, and the maximum values of the swelling degree are in good agreement with NSMs used in preparation parameters. The swelling degrees at 24 h, for all samples in an aqueous medium at different pH, are presented in Figure 10, Figure 11, Figure 12 and Figure 13.

After the first analysis of the experimental data resulting from the study of swelling degree, it can be seen that the values of maximum swelling degree for all samples obtained in acetate buffer solution (pH = 3.6) are superior to those recorded in phosphate-buffered solutions (pH = 7.4 and pH = 6.7) for all preparation parameters. In ABS, pH = 3.6, the swelling rate is between 720 and 436%. For pH = 6.7, the swelling rate is between 315 and 460%, and in PBS, pH = 7.4, it ranges between 425% and 278%.

This behavior is explained as follows: in the acidic environment, most of the amine groups of CS, which have not participated in the crosslinking reactions, have been protonated, passing into the form of ammonium cations (–NH_3_^+^). Therefore, the molecules had a positive charge, and the stronger swelling can be caused by the strong electrostatic repulsion that appears between the macromolecules and leads to an increase in the spaces between the meshes of the polymeric network, that is to say to a retention of water, resulting in a higher swelling degree. If the pH value is higher than 6, the amino groups in CS are less protonated, which limits the swelling degree. Meanwhile, at even higher pH values, like in weakly basic media, most of the amine groups of CS are in the -NH_2_ form (are not protonated). In this case, there are no repulsions, except hydrogen bond-type interactions, which means that the amount of water that penetrates the polymer network is lower, and therefore the swelling rate is also lower. This pH-sensitive characteristic of magnetic nanospheres can be used for the development of pH-sensitive drug delivery systems.

The influence of the polymer/magnetite ratio on the NSM swelling degree was confirmed by the results obtained for the samples NSM-1, NSM-2, NSM-3, NSM-4, and NSM-5, presented in Figure 10.

For the different aqueous media (pH = 3.6, pH = 6.7, and pH = 7.4), the increase in the quantity of magnetite NPs, initially introduced into the particles, leads to a reduction in swelling degree. Thus, the maximum values of swelling are obtained by NSM-1: 708% in ABS and 425% in PBS. On the other hand, for the sample NSM-5, where a higher amount of magnetite is present in the system, the swelling degree is 477% in ABS and 279% in PBS. An explanation for this may be the fact that the polymer (CS) is the direct factor that determines the swelling of the NSMs. So, the increase in the magnetite content leads to a reduction in the quantity of water that can be entrapped by the particles (a more compact network of the NSMs) and, therefore, the decrease in the swelling degree.

According to Figure 11, the decrease in CS concentration solution (0.7%, 0.5%, and 0.3%) leads to a decrease in the values of the water uptake. A possible explanation may be related to a possible higher network crosslinking density.

Figure 12 presents the influence of the stirring speed on the swelling degree. The results show that the stirring speed also had an important impact on the swelling degree of the NSMs, which presented higher values in the case of an increased stirring intensity, namely, 15,000 rpm. It was observed that beyond 15,000 rpm, the swelling degree is lowering, as can be seen for sample NSM-11. This behavior may be attributed to the fact that at a much higher speed, the particle size becomes smaller, and so the NSMs present a more compact network; therefore, the amount of water that penetrates inside becomes lower.

The NH_2_/TPP molar ratio is another parameter influencing the NSMs’ swelling degree, as shown in Figure 13—a logical insight. The increase in the amount of ionic crosslinker (TPP) used in the preparation process (Table 1) leads to a decrease in the swelling degree. This may be due to the formation of a stiffer polymer network with a higher crosslink density. Similar behavior was observed in other studies [40,41].

### 3.8. Loading Efficiency of 5-FU

This type of magnetic nanosphere was developed for use in antitumor therapy. For this reason, our choice is focused on 5-Fluorouracil as a model drug that is commonly used for the treatment of cancer. The results obtained confirm that the quantity of 5-FU included by diffusion is in good agreement with the swelling degree of hybrid nanospheres in aqueous media and with the composition of nanoparticles. Previous studies have reported similar results for different hydrophilic polymer networks [42,43]. The amount of drug loaded into the magnetic nanospheres (5-FUg/g nanoparticles) and the encapsulation efficiency of 5-FU are presented in Table 3.

The drug retained in the magnetic nanoparticles is dependent on the polymer/magnetite ratio. The drug-loading efficiency values increase if the NPs’ magnetic quantity is decreased. The results showed that the highest amount of loaded 5-FU was recorded for particles with the lowest amount of magnetite (NSM-1). Also, the lowest amount of loaded drug was recorded for the particle that had a higher polymer/magnetite ratio (NSM-3). The effect is completely normal due to the space left available to the drug in the magnetic nanosphere being reduced by increasing the amount of magnetite NPs.

### 3.9. Drug-Release Kinetics

The in vitro release experiments were carried out in weakly alkaline environments simulating blood conditions (PBS, pH = 7.4) and in a buffer solution of pH = 6.7. The kinetics of 5-FU release from NSM-1, NSM-2, and NSM-3 samples in PBS (pH = 7.4 and pH = 6.7) are displayed in Figure 14. The release efficiency of the 5-FU in the two media ranged between 35% and 60%, with a slight difference, as expected, due to the protonation CS amine groups. The 5-FU-release efficiency in an aqueous medium at pH = 6.7 was higher compared with that obtained in an aqueous medium at pH = 7.4.

In Figure 14, it can be observed that the 5-FU-release profile from the NSM presents two phases: a first rapid-stage “burst effect” was observed in the first 300 min, followed by a second slow stage (a phase of prolonged release) until the equilibrium characterized by a constant release, when it reaches a maximum 5-FU quantity being loaded. The rapid-release phase corresponds to the release of 5-FU adsorbed on the surface of magnetic nanospheres or encapsulated molecules located near the nanoparticle interface (in the surface layers of the particles). The second release step is attributed to 5-FU trapped in the nanoparticle matrix.

### 3.10. Theoretical Analysis of Drug Release

For the theoretical analysis of 5-FU release, the equation proposed by Peppas et al. described previously (Section 2.3.8) provides us with information about the nature of the diffusion process. Figure 15 highlights the experimental kinetic data of 5-FU-release analysis obtained for the interval 0–300 min, which allowed us to calculate the diffusion exponent. The linearization of this equation (Equation (6)) made it possible to determine the values of the exponent n for the different release profiles. The results obtained are presented in Table 4.

The results highlighted in Table 4 and Figure 15 show that all analyzed magnetic nanospheres (NSM-1, NSM-2, and NSM-3) in the two media (pH = 6.7 and pH = 7.4) presented values of the exponential release factor “*n*” lower than 0.5, which indicates a controlled 5-FU release from NSMs appropriate to Fickian diffusion.

## 4. Conclusions

Superparamagnetic hybrid nanoparticles intended for the target transport of antitumor drugs were obtained using the method of double crosslinking in an inverse emulsion via the inclusion of magnetite nanoparticles in a matrix of chitosan. To reduce the quantity of glutaric aldehyde, considered inadequate for obtaining biomaterials due to its toxicity, tripolyphosphate (ionic crosslinker) was used as a quantitatively dominating crosslinker.

The morphology of the hybrid nanoparticles obtained (diameter and dimensional polydispersity), evaluated by light scattering and by scanning electron microscopy, is dependent on the parameters of the production process.

The average diameter of the particles decreases by decreasing the rate of magnetite and the concentration of the chitosan solution, by increasing the speed of agitation of the system, and by increasing the concentration of the crosslinker ratio (both ionic and covalent).

The swelling behavior of hybrid nanoparticles in liquids at different pH values also depends on the mentioned parameters.

The ability to include and release 5-Fluorouracil (a model antitumor drug) correlates very well with the swelling behavior; it increases with an increasing degree of nanoparticle swelling. The mechanism of transport and release of the drug, evaluated based on the Peppas equation, indicates a behavior close to Fickian behavior.

## Figures and Tables

**Figure 1 polymers-15-04493-f001:**
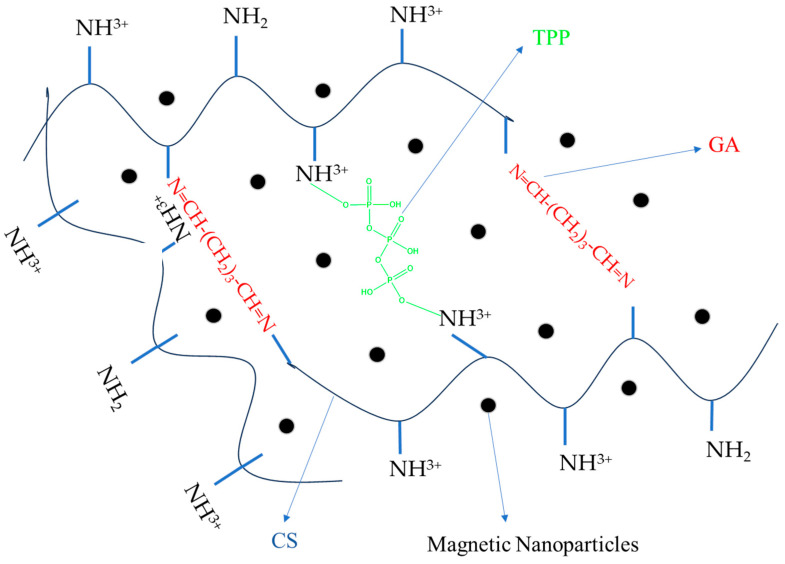
Schematic route of NSMs based on CS containing magnetic nanoparticles (ionic and covalent crosslinking reaction between CS and crosslinking agents).

**Figure 2 polymers-15-04493-f002:**
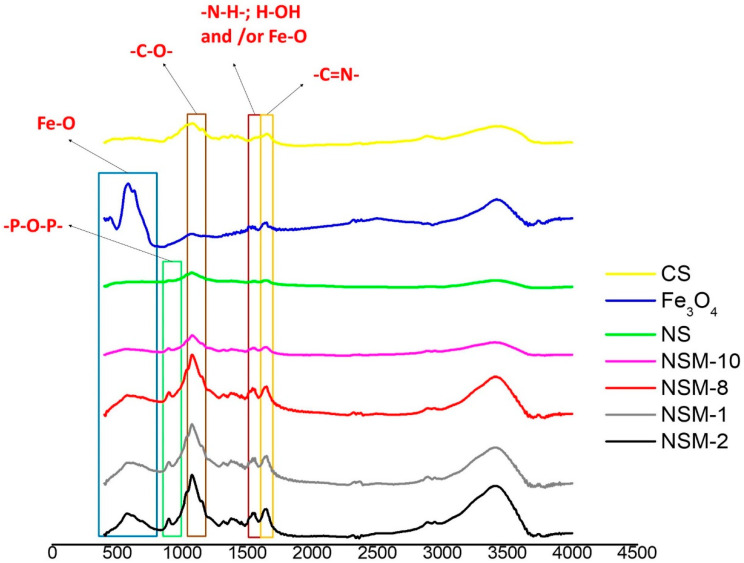
FT-IR spectra for CS, magnetite (Fe_3_O_4_), NS, NSM-1, NSM-2, NSM-8, and NSM-10.

**Figure 3 polymers-15-04493-f003:**
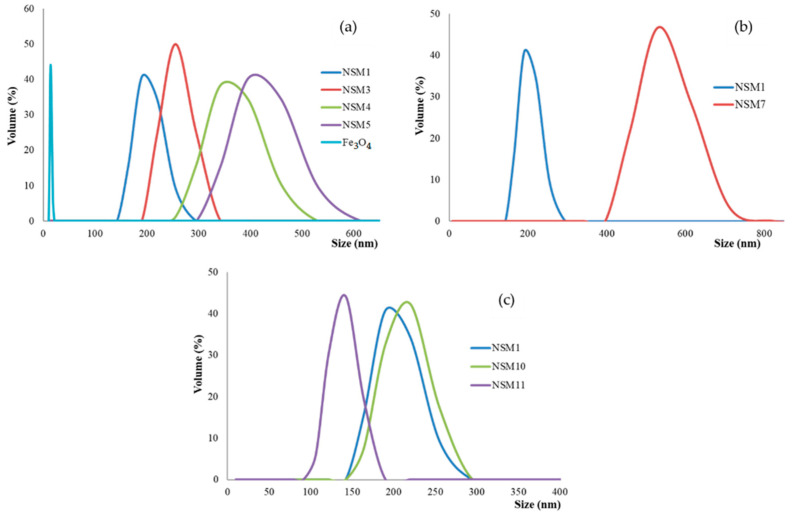
Dimensional distribution curves of: (**a**): Nanosphere samples at different ratios: NSM-1(CS/Fe_3_O_4_:1/0.5; 15,000 rpm), NSM-3(CS/Fe_3_O_4_: 1/1; 15,000 rpm), NSM-4 (CS/Fe_3_O_4_:1/1.2; 15,000 rpm), and NSM-5 (CS/Fe_3_O_4_: 1/1.7; 15,000 rpm); (**b**): Nanosphere samples at different polymer concentrations: NSM-1 (CS: 0.5%) and NSM-7 (CS: 0.7%); (**c**): Nanosphere samples at different stirring speeds: NSM-1(CS/Fe_3_O_4_: 1/0.5; 15,000 rpm), NSM-10 (CS/Fe_3_O_4_: 1/0.5; 12,000 rpm), and NSM-11(CS/Fe_3_O_4_: 1/0.5; 12,000 rpm). The correlation dimension/preparation parameters of the NSMs are presented in (**a**–**c**). (**a**) highlights that a high quantity of magnetite leads to hybrid nanospheres of a larger size, thus confirming that the size of particles depends on the CS/Fe_3_O_4_ wt ratio. In the case of samples NSM-1, NSM-3, NSM-4, and NSM-5, we note that the particle size increases from 190 to 459 nm, which confirms the influence of the CS/Fe_3_O_4_ wt ratio. This result also highlights that the magnetic nanoparticles are present in and around (on the surface) of magnetic nanospheres.

**Figure 4 polymers-15-04493-f004:**
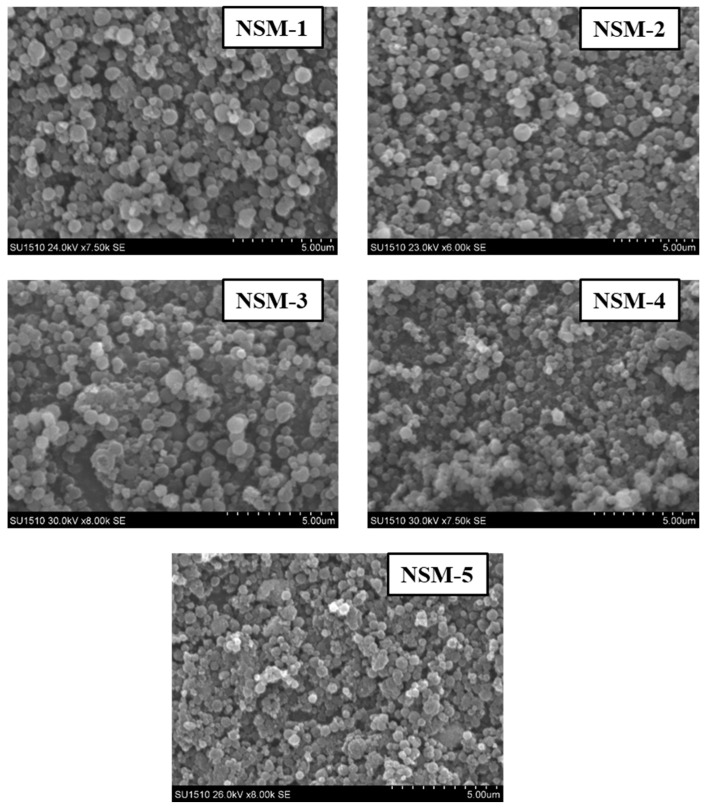
SEM images of NSMs with different CS/Fe_3_O_4_ weight ratios.

**Figure 5 polymers-15-04493-f005:**
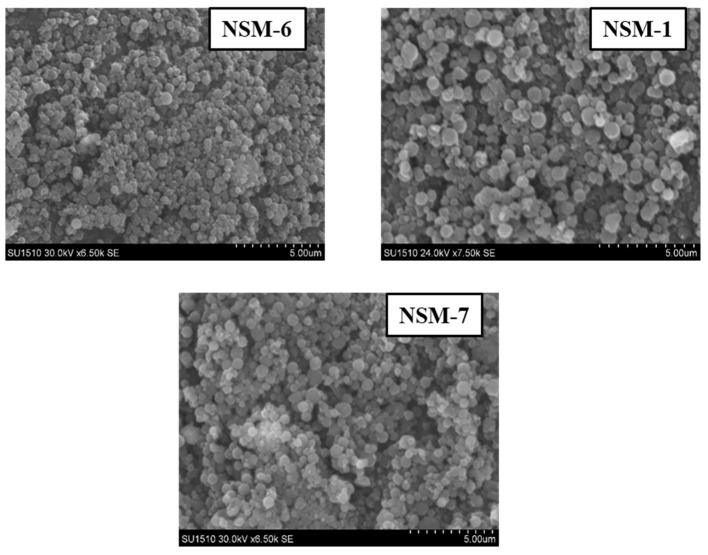
SEM images of NSMs with different concentrations of chitosan solution.

**Figure 6 polymers-15-04493-f006:**
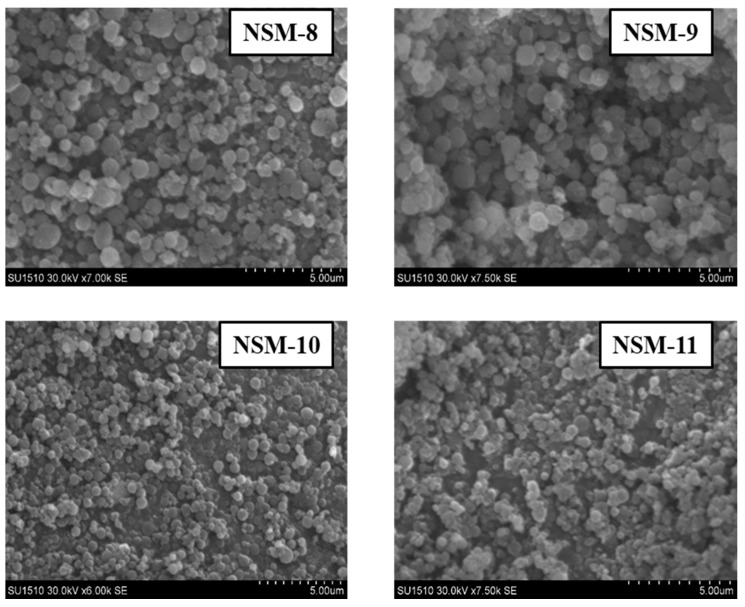
SEM images of NSMs obtained with different stirring speeds.

**Figure 7 polymers-15-04493-f007:**
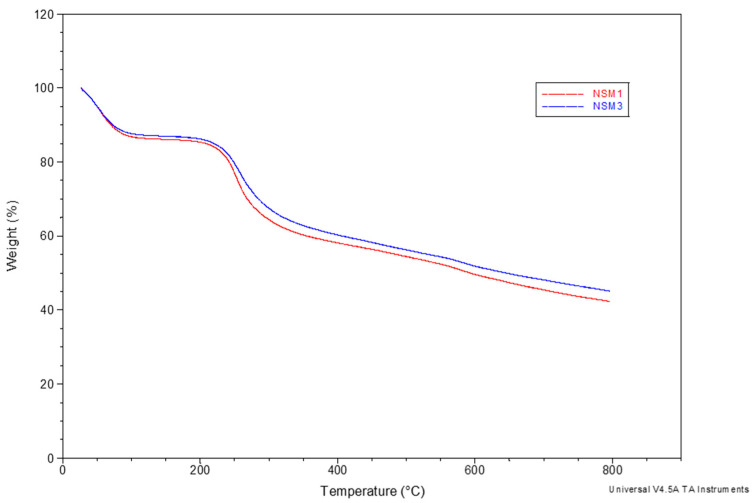
Thermogravimetric curves for NSM-1 and NSM-3.

**Figure 8 polymers-15-04493-f008:**
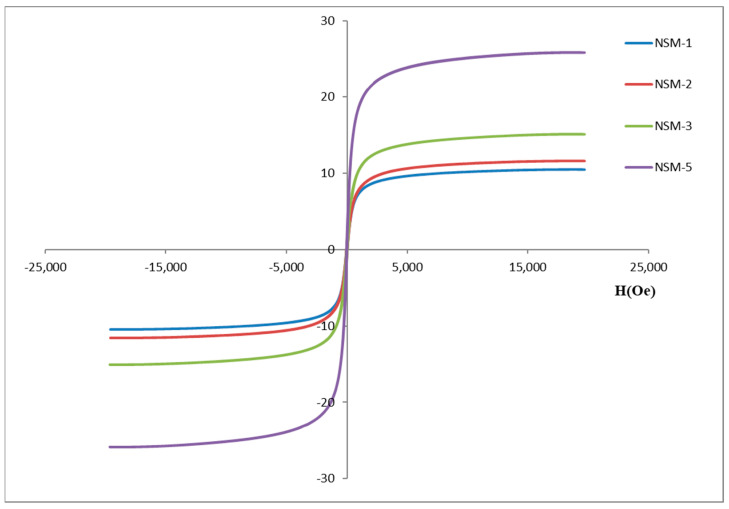
The magnetization curves for NSM-1, NSM-2, NSM-3, and NSM-5 samples.

**Figure 9 polymers-15-04493-f009:**
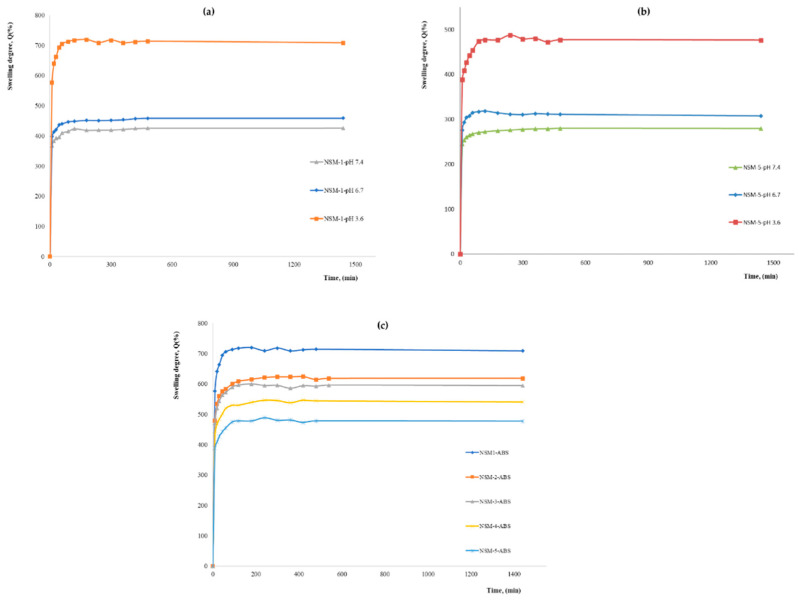
The swelling kinetics curves of magnetic nanospheres in different aqueous solutions: (**a**) kinetics of NSM-1 sample in pH = 3.6, pH = 6.7, and pH = 7.4; (**b**) kinetics of NSM-5 sample in pH = 3.6, pH = 6.7, and pH = 7.4; (**c**) kinetics of NSM-1, NSM-2, NSM-3, NSM-4, and NSM-5 samples in ABS (pH = 3.6).

**Figure 10 polymers-15-04493-f010:**
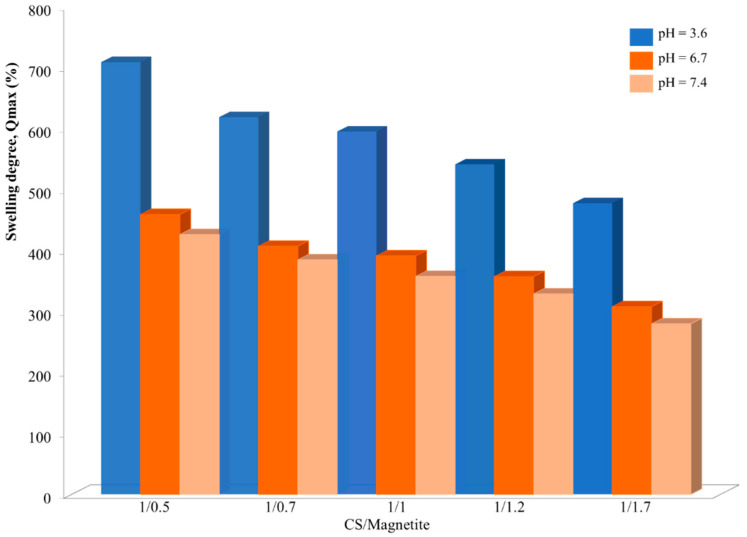
The influence of polymer/magnetite ratio on swelling degree, after 24 h, in solutions with different pHs (pH = 3.6, pH = 6.7, and pH = 7.4).

**Figure 11 polymers-15-04493-f011:**
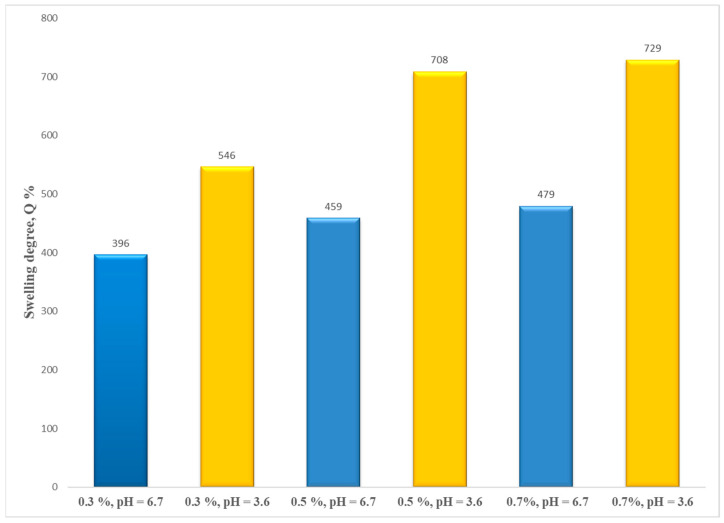
The influence of polymer concentration on the swelling degree.

**Figure 12 polymers-15-04493-f012:**
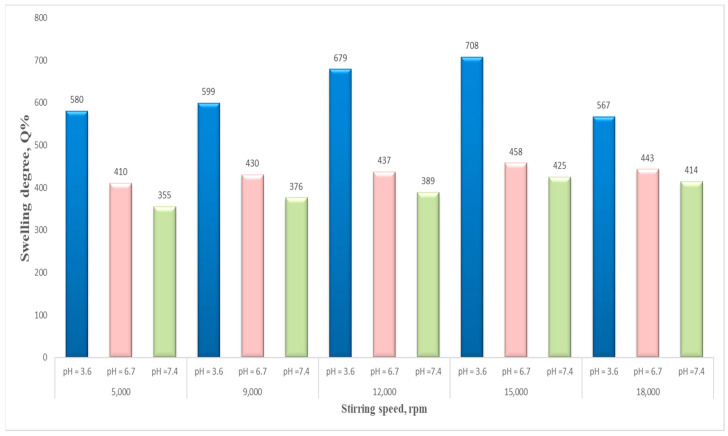
Stirring speed influence on swelling degree (at 24 h) in different aqueous solutions (pH = 3.6, pH = 6.7, and pH = 7.4) (NSM-6: 5000 rpm, NSM-7: 9000 rpm, NSM-8: 12,000 rpm, NSM-10: 15,000 rpm, and NSM-9: 18,000 rpm).

**Figure 13 polymers-15-04493-f013:**
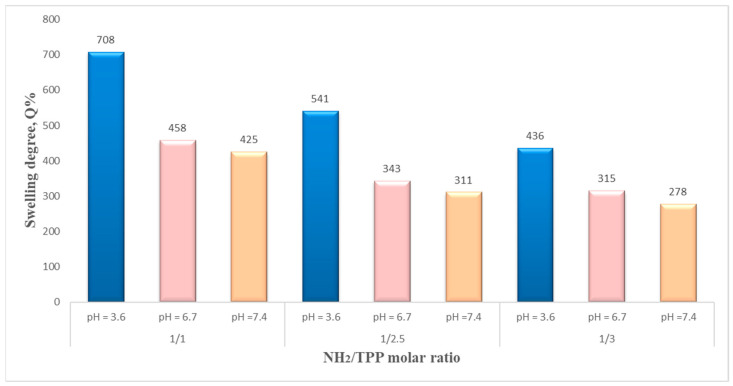
The molar ratio CS/TPP influences the swelling degree (NSM-1: CS/TPP:1/2, NSM-12. CS/TPP:1/2.5, and NSM-13. CS/TPP:1/3).

**Figure 14 polymers-15-04493-f014:**
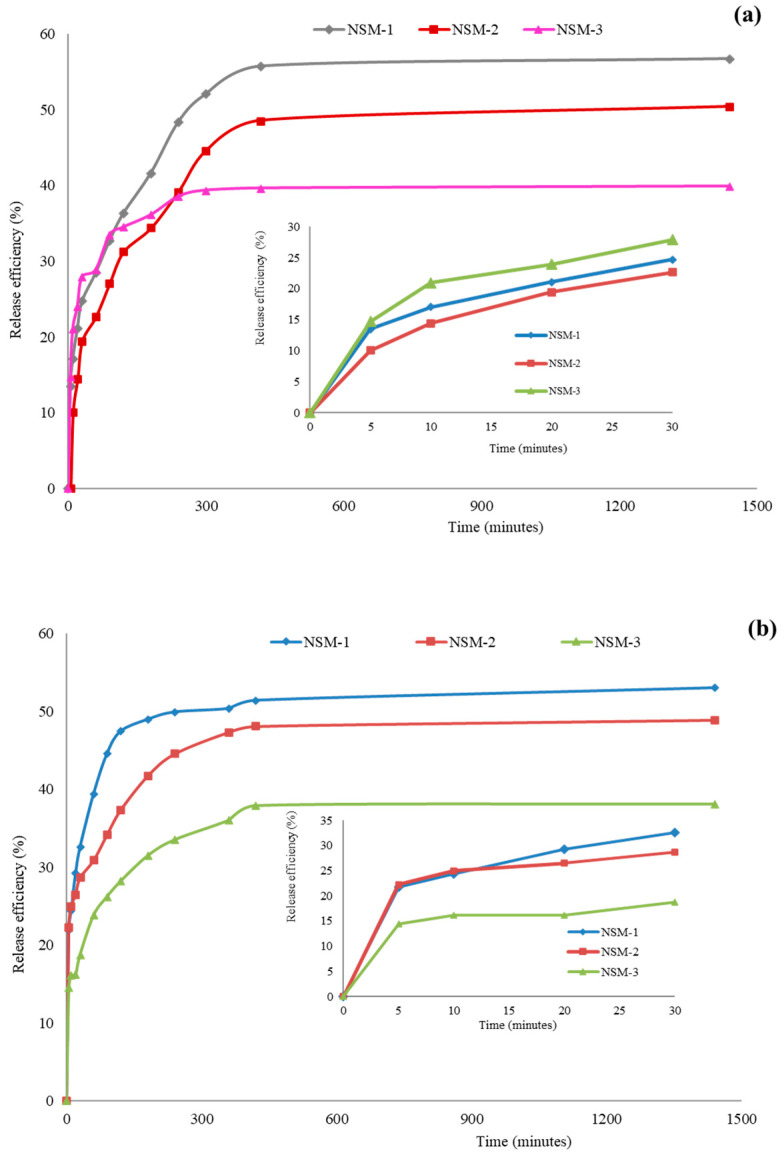
In vitro release kinetics of 5-FU from hybrid nanoparticles with a zoom insert of release kinetics between 0 and 30 min for samples NSM-1, NSM-2, and NSM-3. (**a**) In buffer solution (pH = 6.7); (**b**) In phosphate-buffered solution (pH = 7.4).

**Figure 15 polymers-15-04493-f015:**
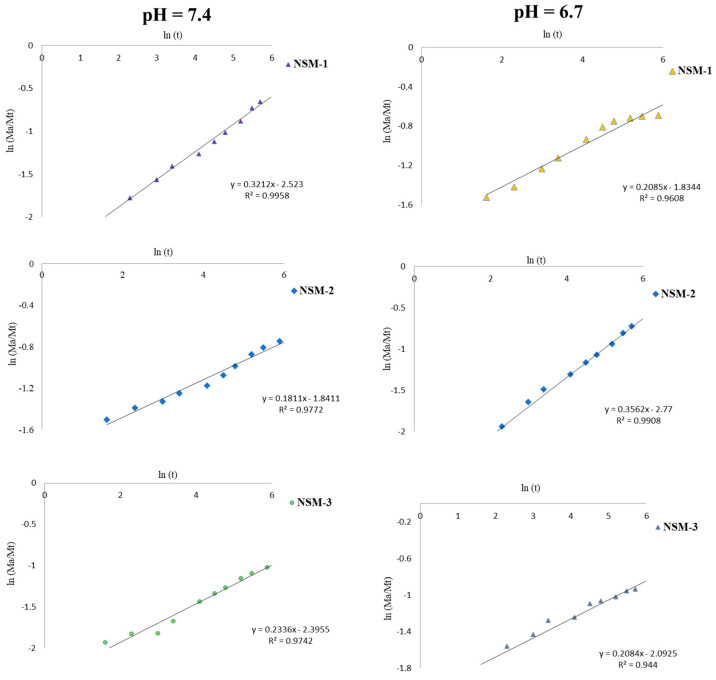
The theoretical curves according to the Korsmeyer–Peppas model for NSM-1, NSM-2, and NSM-3 samples.

**Table 1 polymers-15-04493-t001:** Experimental program and codification of synthesized samples.

Sample Code	CS/Fe_3_O_4_ Ratio (*w*/*w*)	% CS (*w*/*v*)	Stirring Speed (rpm)	CS/TPP Ratio (mol/mol)
NSM-1	1/0.5	0.5	15,000	1/2
NSM-2	1/0.7	0.5	15,000	1/2
NSM-3	1/1	0.5	15,000	1/2
NSM-4	1/1.2	0.5	15,000	1/2
NSM-5	1/1.7	0.5	15,000	1/2
NSM-6	1/0.5	0.3	5000	1/2
NSM-7	1/0.5	0.7	9000	1/2
NSM-8	1/0.5	0.5	12,000	1/2
NSM-9	1/0.5	0.5	18,000	1/2
NSM-10	1/0.5	0.5	15,000	1/2
NSM-11	1/0.5	0.5	5000	1/2
NSM-12	1/0.5	0.5	9000	1/2.5
NSM-13	1/0.5	0.5	12,000	1/3

**Table 2 polymers-15-04493-t002:** NSMs’ Zeta potential measurements.

Samples	Zeta Potential, (mV)
Fe_3_O_4_	−25.6
NSM-2	−14.3
NSM-3	−9.37
NSM-5	−19.7
NSM-6	−14.4
NSM-7	−7.05
NSM-8	−8.15
NSM-9	−8.4
NSM-11	−8.65

**Table 3 polymers-15-04493-t003:** The 5-FU encapsulation efficiency.

Sample Code	Loaded 5-FU mg/mg Nanospheres	Loading Efficiency
NSM-1	0.201	26.82
NSM-2	0.192	25.6
NSM-3	0.1455	19.4

**Table 4 polymers-15-04493-t004:** Korsmeyer–Peppas parameters.

Sample	Exponential Factor (*n*)	
	pH = 6.7	pH = 7.4
NSM-1	0.3212	0.2085
NSM-2	0.3562	0.1811
NSM-3	0.2084	0.2336

## Data Availability

Data are contained within the article.

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
