# Peer review of "Superparamagnetic Hybrid Nanospheres Based on Chitosan Obtained by Double Crosslinking in a Reverse Emulsion for Cancer Treatment"

_polymers, 2023, doi:10.3390/polym15234493_

Round 1
Reviewer 1 Report
Comments and Suggestions for Authors
1. In Figure 2, the number of wavenumbers marked on the infrared spectra should be taken as an integer.
2. In a medium with pH 6.7, the magnetic nanospheres (NSM-1) loaded with 5-FU released up to 55% of 5-FU. Over time, there was no longer a release of 5-FU in Figure 14a. Why? From Figure 14b, it can be seen that in a buffer solution with pH 7.4, the release of 5-FU reached 35% to 50%. This means that the drug 5-FU leaked out in the normal internal environment. The magnetic nanoparticles loaded with the drug did not achieve the expected results.
3. On page 9, lines 357-359, the authors stated that “Particle size measurements were performed via dynamic light scattering and confirmed that magnetic NPs have an average diameter of 14 nm, respectively for the size of the hybrid nanosphere varied between 190 and 531 nm (Figure 3)”. I suggest that the authors consider how the drug loaded hybrid magnetic nanoparticles can be used for in vivo therapy. If intravenous injection is chosen, the particle size of magnetic nanoparticles should preferably not exceed 200 nm. In addition, the authors should also demonstrate that the drug-loaded hybrid nanoparticles do not agglomerate.
4. The authors provided data and graphics for Figure 15 at the end of the manuscript, but did not mention and discuss the figure in the main text.
5. The authors should also carefully examine the manuscript. For example, for Fe3O4, 3, and 4 should be changed to subscripts. On page 9, lines 340-342, the wavenumber unit of the infrared spectrum is "cm-1", and "-1" should be changed to superscript.
Comments on the Quality of English LanguageEnglish language can still be improved.
Reviewer 2 Report
Comments and Suggestions for Authors
file

Round 2
Reviewer 1 Report
Comments and Suggestions for Authors
The authors have answered my questions and made appropriate revisions to their manuscript. Therefore, I recommend publishing this revised manuscript.
Reviewer 2 Report
Comments and Suggestions for Authors
Accept